# Do pregnant African women exercise? A meta-analysis

**Yohannes Fikadu Geda**[1]*, **Seid Jemal Mohammed**[1], **Tamirat Melis Berhe**[1], **Samuel Ejeta Chibsa**[2], **Tadesse Sahle**[1], **Yirgalem Yosef Lamiso**[1], **Kenzudin Assfa Mossa**[1], **Molalegn Mesele Gesese**[3]

**1** Wolkite University, Wolkite, Ethiopia, **2** Mettu University, Mettu, Ethiopia, **3** Wolaita Sodo University, Wolaita Sodo, Ethiopia

* nechsar@gmail.com

## Abstract

### Introduction

Antenatal exercise can reduce gestational weight gain, backache; pregnancy induced medical disorders, caesarean section rates, and improves pregnancy outcomes. American College of Obstetrics and Gynecology (ACOG) recommends prenatal exercise, which is associated with minimal risk and has been shown to be beneficial for pregnancy outcomes, although some exercise routines may need to be modified. Consequently, this meta-analysis is intended to verify the pooled practice of antenatal exercise in Africa using available primary articles.

### Methods

Genuine search of the research articles was done via PubMed, Scopes, Cochrane library, the Web of Science; free Google databases search engines, Google Scholar, and Science Direct databases. Published and unpublished articles were searched and screened for inclusion in the final analysis and Studies without sound methodologies, and review and meta-analysis were not included in this analysis. The Newcastle–Ottawa scale was used to assess the risk of bias. If heterogeneity exceeded 40%, the random effect method was used; otherwise, the fixed-effect method was used. Meta-analysis was conducted using STATA version 14.0 software. Publication bias was checked by funnel plot and Egger test.

### Results

This review analyzed data from 2880 women on antenatal care contact from different primary studies. The overall pooled effect estimate of antenatal exercise in Africa was 34.50 (32.63–36.37). In the subgroup analysis for pooled antenatal exercise practice by country, it was 34.24 (31.41–37.08) in Ethiopia and 37.64(34.63–40.65) in Nigeria.

### Conclusion

The overall pooled effect estimate of antenatal exercise in Africa was low compared to other continent. As it was recommended by ACOG antenatal exercise to every patient in the

**Data Availability Statement:** All relevant data are within the paper and its Supporting information files.

**Funding:** The authors received no specific funding for this work.

**Competing interests:** The authors have declared that no competing interests exist.

**Abbreviations:** ACOG, American College of Obstetrics and Gynecology; CI, Confidence Interval; EI, Effect Estimate; NOS, Newcastle-Ottawa Scale; PRISMA, Preferred Reporting Items of Systematic Reviews and Meta-Analysis; WHO, World Health Organization.

absence of contraindications, it should be encouraged by professionals providing antenatal care service.

## 1. Introduction

Exercise is a physical activity requiring physical effort, carried out to sustain or improve health and fitness which consists of planned, structured, and repetitive body movements [1, 2]. Whereas antenatal exercises is an exercises conducted to improve the physical and psychological well-being of a pregnant mother and preventing pregnancy-induced medical disorders using various physical activities [3, 4]. If complications are prevented with antenatal exercise and other measures of health promotion pregnancy is a natural condition [5].

A systematic review shows that a large number of women were obtained weight than is recommended during pregnancy; which is a risk factor of the complications [6]. Antenatal exercise is always important to both the parturient and the perinatal during the course of pregnancy, labor/delivery and in postpartum recovery if it's conducted with best available medical recommendations [7, 8].

Antenatal exercise can reduce gestational weight gain, backache; pregnancy induced medical disorders, caesarean section rates, and improves pregnancy outcomes [2, 9–14]. There is a number of tangible evidence that antenatal exercise practice help to reduce excessive weight gain during pregnancy [6, 15]. This exercise can also be an essential factor by preventing postpartum depression and improving sleep quality [3, 16, 17]. Lower back and pelvic pain in pregnant women have a prevalence of 50% which can be skipped by low to moderate exercises [18].

Improved neonatal outcome is predicated on proper antenatal exercise [19]. For instance gestational diabetes mellitus can be controlled by antenatal exercise, which in turn reduces trauma of the new born due to Macrosomia during delivery [20]. In the other hand moderate physical exercise throughout pregnancy may increases fetal cardiovascular development [4]. Beside this pre-pregnancy physical activity had no negative effects on preterm birth [19, 21].

American College of Obstetrics and Gynecology (ACOG) [1] recommended that exercise performed for at least 20 to 30 minutes at least three times a week can improve overall fitness considering anatomic and physiologic changes. Physical exercise such as low to moderate aerobics, walking, swimming, dancing, stationary cycling and stretching can be listed as a safe during pregnancy in the absence of medical or obstetrical complications [10, 22]. But obstetric and medical conditions contraindicated to antenatal exercise, vigorous physical activity under dehydration and exercise in the supine position need to be warned [1].

Sedentary life style during pregnancy might result in compromised physical fitness, unnecessary maternal weight gain, varicose veins, lower backache and poor psychological preparedness [9, 23]. In addition pain around the pelvic might be associated with a minimized in regular physical activity during pregnancy [24]. Similarly pregnancy induced medical disorders may result in fetal complications such as growth restriction, oligo-hydramnios, placental abruption, preterm birth and perinatal death [13].

Given the overwhelming importance of antenatal exercise, available data in Africa were not too much. Moreover there is no study which combines all available studies to have a single pooled effect size. Consequently, this meta-analysis is intended to verify the pooled practice of antenatal exercise in Africa using available primary articles. The result and conclusion of this study will bring rigor information which can be abele to be used by program planners, other researchers and policy developers to obtain qualified health care service delivery. As well, this study can be used by health professionals who use evidence-based practices to deliver services.

## 2. Methods and materials

### 2.1 Study design and setting

The authors have assessed the PROSPERO database (https://www.crd.york.ac.uk/PROSPERO/) for all published or ongoing researches available related to the title to avoid any further duplication. Accordingly the result brought that there were no ongoing or published articles in the area of this title. Therefore, this meta-analysis was registered in the PROSPERO database with an identification number of CRD42023388265 on 15/01/2023. It was conducted to verify the pooled antenatal exercise in Africa. Scientific consistency was formulated by using PRISMA checklist [25].

### 2.2 Information source

A systematic and authentic search for the research papers was carried out using the databases listed below. Scopes, PubMed, the Web of Science, Cochrane library, Google Scholar, and Science Direct search engines were included in the review. We have used the keywords (((((("Antenatal Exercise/exercise"[Mesh] OR "Exercise during Pregnancy/antenatal"[Mesh])) AND "Risk Factors"[Mesh]) AND "all African countries interchangeably"[Mesh])) OR "Africa"[Mesh:NoExp]

The search was carried out using the following search keywords: "AND" and "OR" Boolean operators individually and in combination with one another. In addition, reference lists for all included studies were also consulted to find additional studies that could have been omitted by the research strategy. The institutions with institutional repositories were exhaustively searched for master's theses and PhD dissertations. In each nation, institutions with repositories were found, and the specified title was then looked for at those institutions. The search for all the studies took place from October 10 to December 20, 2022, without limiting the publication dates of the literature.

### 2.3 Eligibility criteria

**2.3.1 Inclusion criteria.**　Published articles in peer reviewed journals, and unpublished articles from institutional data base conducted in Africa which have a result and conclusion on antenatal exercise were included in this study. Published and unpublished researches were searched and screened for inclusion in the final model of the analysis. This study included all observational cross-sectional studies conducted in any African country, which reports magnitude of antenatal exercise. All researches that were published, master's thesis found in institutional repositories, and PhD dissertation accessed from the repositories till the final date of data analysis and submission of this manuscript to this journal were included in accordance with these criteria.

**2.3.2 Exclusion criteria.**　Studies that did not have proven methods were not included in this analysis. Articles that do not contain comprehensive information of importance for analysis and case reports were excluded from the study. Overlapping results in studies and inconsistent measures of outcome variables were excluded from the final analysis. Moreover studies published in a language other than English were excluded (Fig 1).

### 2.4 Operational definition

**2.4.1 Antenatal exercise.**　When a pregnant woman engaged in any type of exercise for 20 to 30 minutes at least three times per week, primary studies indicated "yes" to antenatal exercise; otherwise, they indicated "no" [1].

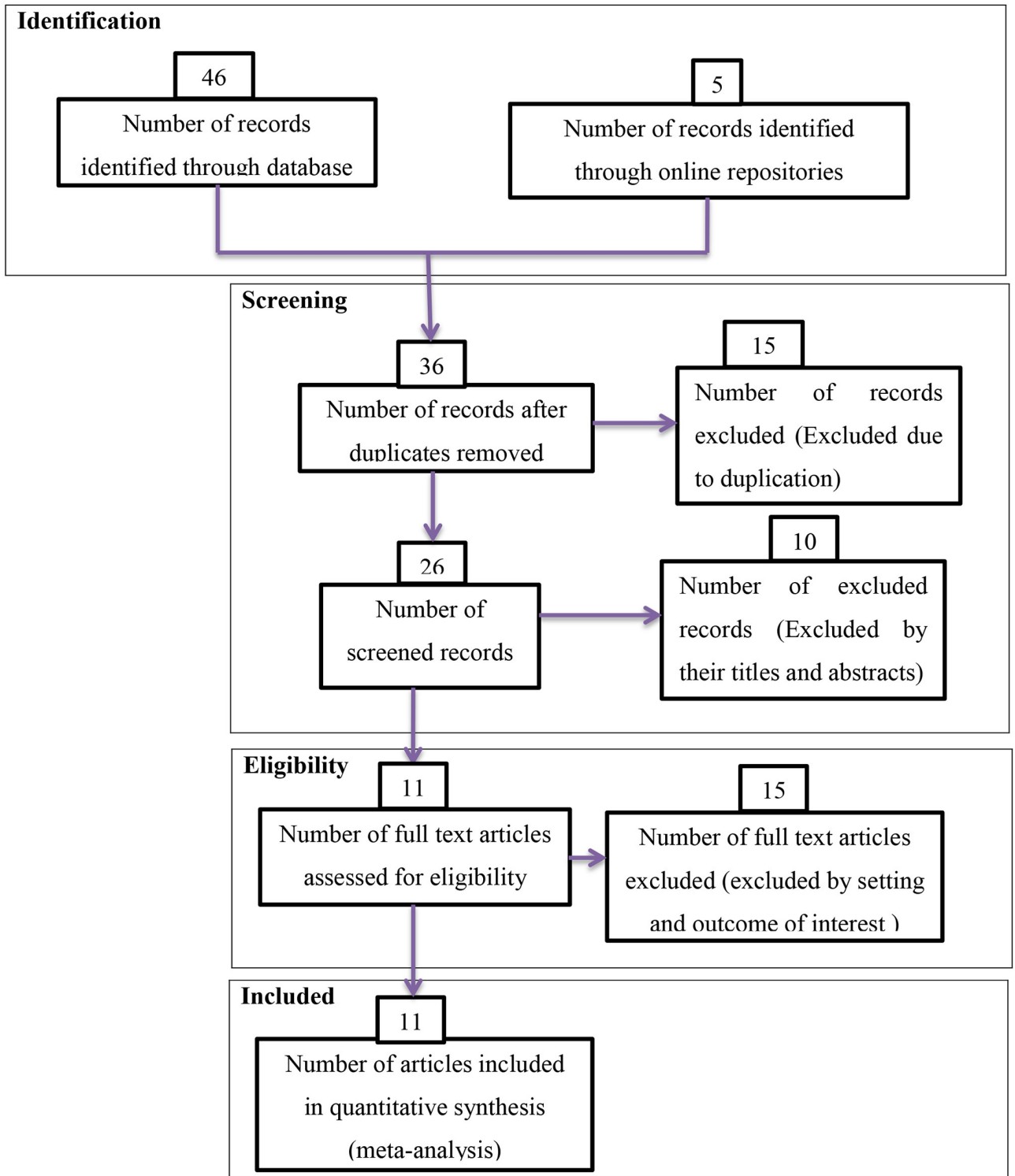

**Fig 1. PRISMA flow diagrams of included studies in the meta-analysis on antenatal exercise in Africa, 2022.**

## 2.5 Study selection

At the outset of our research, we found 51 studies, 15 of which were skipped because of overlap, and the other 36 studies were selected for eligibility. From 36 studies 10 were excluded by highlight review on their abstracts and 26 studies assessed for full text from this 15 studies excluded because of not relevant to the current review and the remaining 11 studies were included in the final meta-analysis of this study (Fig 1).

## 2.6 Quality assessment and data extraction

The baseline quality of the research articles included was assessed using the Newcastle-Ottawa scale (NOS). NOS were used to assess the quality of articles with a bunch of criteria's in in this study (S2 Table). Data from this study were extracted by the two authors (YFG and SJM) using a standardized checklist for extracting data on an Excel sheet.

This meta-analysis uses the PRISMA flowchart to differentiate and select items of significance to the analysis. Initially, duplicate types of studies were not included using the Endnote version X8.1 referencing tool. Articles were excluded by adding highlights by going through their titles and abstracts before evaluating the entire text. Full-text studies or research results have been evaluated for other studies. Based on the aforementioned eligibility criteria; items have been assessed for eligibility.

Data were extracted using the standardized data extraction tool in considering the name of the first author, publication year, country of study, author's affiliation, sample size, magnitude of antenatal exercise and their 95% confidence interval (Table 1). All articles were independently verified by the two authors (YFG and SJM). Where disagreements have occurred, the articles have been reviewed by one of the authors (TMB) and used as final mediation and admissibility decision.

## 2.7 Data synthesis and analysis

The analysis of this meta-analysis was conducted by STATA version 14.0. Quantitative reviews were conducted to determine the overall pooled antenatal exercise in Africa. The degree of heterogeneity between the included studies was evaluated by determining the p-values of $I^2$-test statistics. $I^2$ test statistics scores of 0, 25, 50, and 75% were taken as no, low, moderate, and high degrees of heterogeneity, respectively [26]. Due to the observed moderate heterogeneity across studies, we used a random effect model to assess pooled estimate. Publication bias was checked by funnel plot. A p-value of less than 0.05 was used as cutoff point for statistical

**Table 1. Descriptive summary of included articles to antenatal exercise in Africa, 2022.**

| Authors | Year | Country | Affiliation of the authors | Sample size |
|---|---|---|---|---|
| A.Nkhata et al. [27] | 2015 | Zambia | University of Zambia | 300 |
| Addis et al. [28] | 2022 | Ethiopia | University of Gondar | 333 |
| Bayisa et al. [29] | 2022 | Ethiopia | Wollega University | 475 |
| Beyene et al. [22] | 2022 | Ethiopia | Arba Minch University | 410 |
| DRC.C et al. [30] | 2017 | Nigeria | University of Nigeria | 204 |
| J.I.B et al. [7] | 2018 | Nigeria | Nnamdi Azikiwe University | 115 |
| Janakiraman et al. [31] | 2021 | Ethiopia | Mekelle University | 349 |
| Mbada et al. [32] | 2014 | Nigeria | Obafemi Awolowo University | 189 |
| Mervat et al. [33] | 2016 | Egypt | Cairo University | 100 |
| Ngayimbesha et al. [5] | 2018 | Burundi | Université du Burundi | 150 |
| Sitot et al. [9] | 2022 | Ethiopia | Mekelle University | 255 |

significance of publication bias. Egger test was done and verified that there was no small-study effects.

## 3. Results

### 3.1 Selection and characterization of included studies

Eleven articles [5, 7, 9, 22, 27–33] were included in this meta-analysis and it was summarized in Table 1. All prevalence studies were included in accordance with the eligibility criteria with the sample size ranging from 115 in Nigeria [7] to 475 in Ethiopia [29] (Table 1).

In relation to geographical region, one study from East-central Africa [5], one study from Northeastern part of Africa [33], five studies from Horn of Africa [9, 22, 28, 29, 31], three studies from West Africa [7, 30, 32] and one study from south-central Africa [27] (Table 1).

This meta-analysis was analyzed data from 2880 women on antenatal care contact to estimate the pooled antenatal exercise in Africa. This meta-analysis included all articles that met the requirements for inclusion (Table 1).

### 3.2 Publication bias

Bias among the included studies was checked by funnel plot at a 5% significant level. The funnel plot was symmetry, and showed no statistical significance for the presence of publication bias for each study. Egger test was done and verified that there was no small-study effects with P = 0.17 (Fig 2).

### 3.3 Antenatal exercise practice in Africa

Eligible magnitude studies were included in the final meta-analysis. Due to observed moderate heterogeneity among the studies random effect model were employed. The overall pooled effect estimate of antenatal exercise in Africa was 34.50 with 95% confidence interval of 32.63 to 36.37 (Fig 3).

### 3.4 Subgroup analysis of antenatal exercise

Subgroup analysis for pooled antenatal exercise practice by country was conducted. Unfortunately more than two studies were available in Ethiopia and Nigeria. Accordingly antenatal exercise in Ethiopia was 34.24 with 95% confidence interval of 31.41 to 37.08; whereas antenatal exercise in Nigeria was 37.64 with 95% confidence interval of 34.63 to 40.65 (Fig 4).

## 4. Discussion

ACOG recommends antenatal exercise, which is associated with minimal risks and have been shown to benefit most women, although some modification to exercise routines may be necessary [1]. In accordance with ACOGs recommendation antenatal exercise in Africa was not satisfactorily practiced even if the actual figure antenatal exercise was not known. With this concept in mind this study assess all available studies (published or unpublished) based on the settled eligibility criteria.

Studies conducted on all 54 countries of Africa according to united nation, they were extremely reviewed as it was stated on the search strategies. Few studies were existed on antenatal exercise in Africa, which was limited to some list of countries.

Available primary studies were included in this study irrespective of publication or study year. By default the included studies were conducted from 2014 to 2022. There were no studies conducted on antenatal exercise practice in Africa. All studies in Africa which reports magnitude of antenatal exercise practice were taken as a part of this study. The limited data available

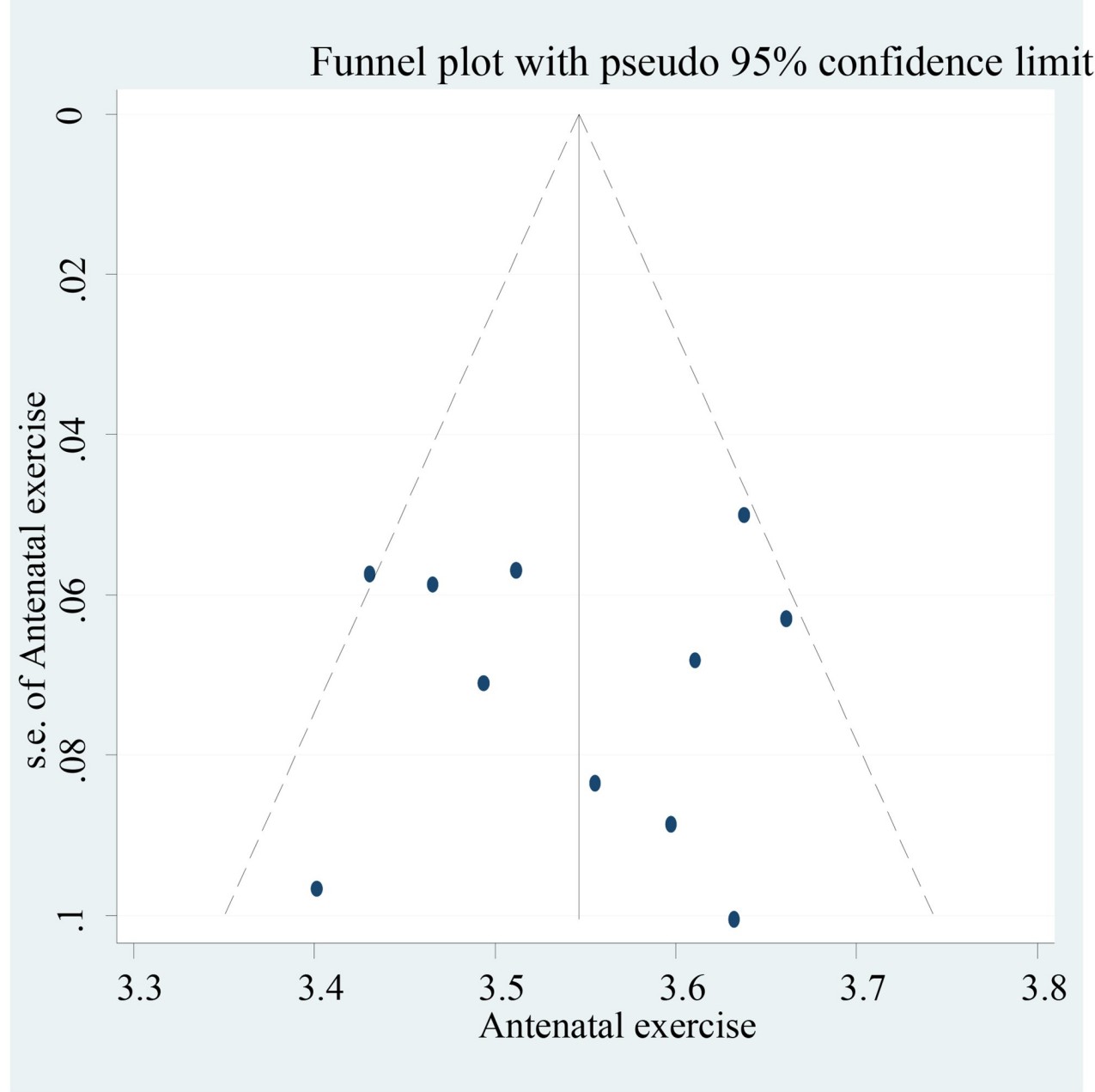

**Fig 2. Funnel plot for studies on antenatal exercise in Africa, 2022.**

suggests that, compared to the Western world, antenatal exercises in Africa do not adhere to the recommendations [1] for exercise during pregnancy.

This study revealed that overall pooled effect estimate of antenatal exercise in Africa was 34.50 with 95% confidence interval of 32.63 to 36.37. In the other hand a study conducted in North America shows about 46% women engage in exercise during pregnancy [16]. This difference might be due to the communities in North America had better opportunities to learn and practice on the effects of antenatal exercise on health outcomes. In the other hand this

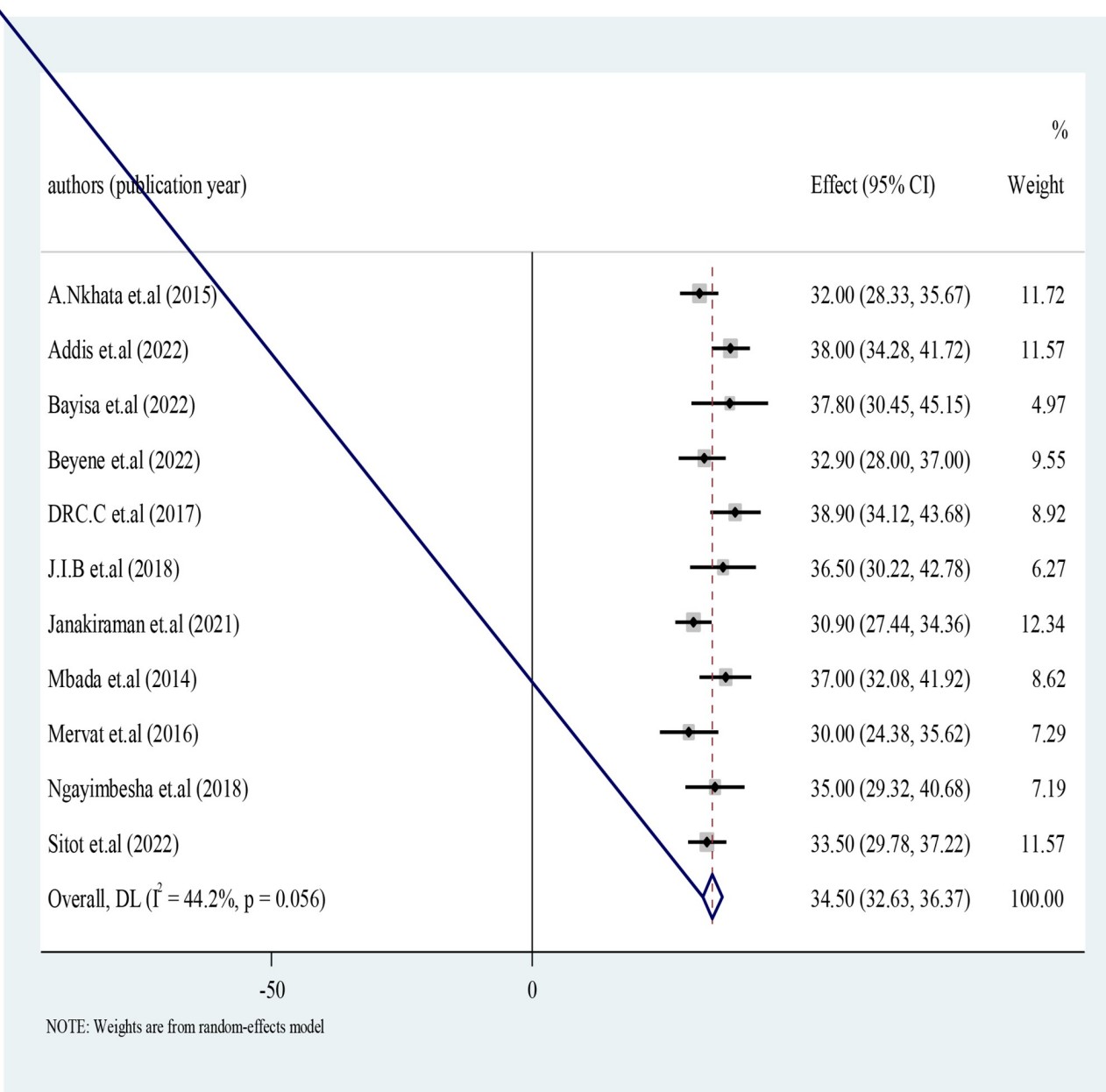

**Fig 3. Forest plot for antenatal exercise in Africa, 2022.**

difference might also be due to the easily available infrastructures of health care in North America expose them to practice antenatal exercise.

On the sub group analysis by country, antenatal exercise in Ethiopia was 34.24 with 95% confidence interval of 31.41 to 37.08. Conversely a study conducted in India was lower than the stated value of antenatal exercise in Ethiopia [19]. This difference might be due older time of study conducted, which was 2015. Since the time goes on modernization by itself brings better understanding in the health care. In the other hand it might be due to pregnant women's in Ethiopia had better understands the health benefit of antenatal exercise.

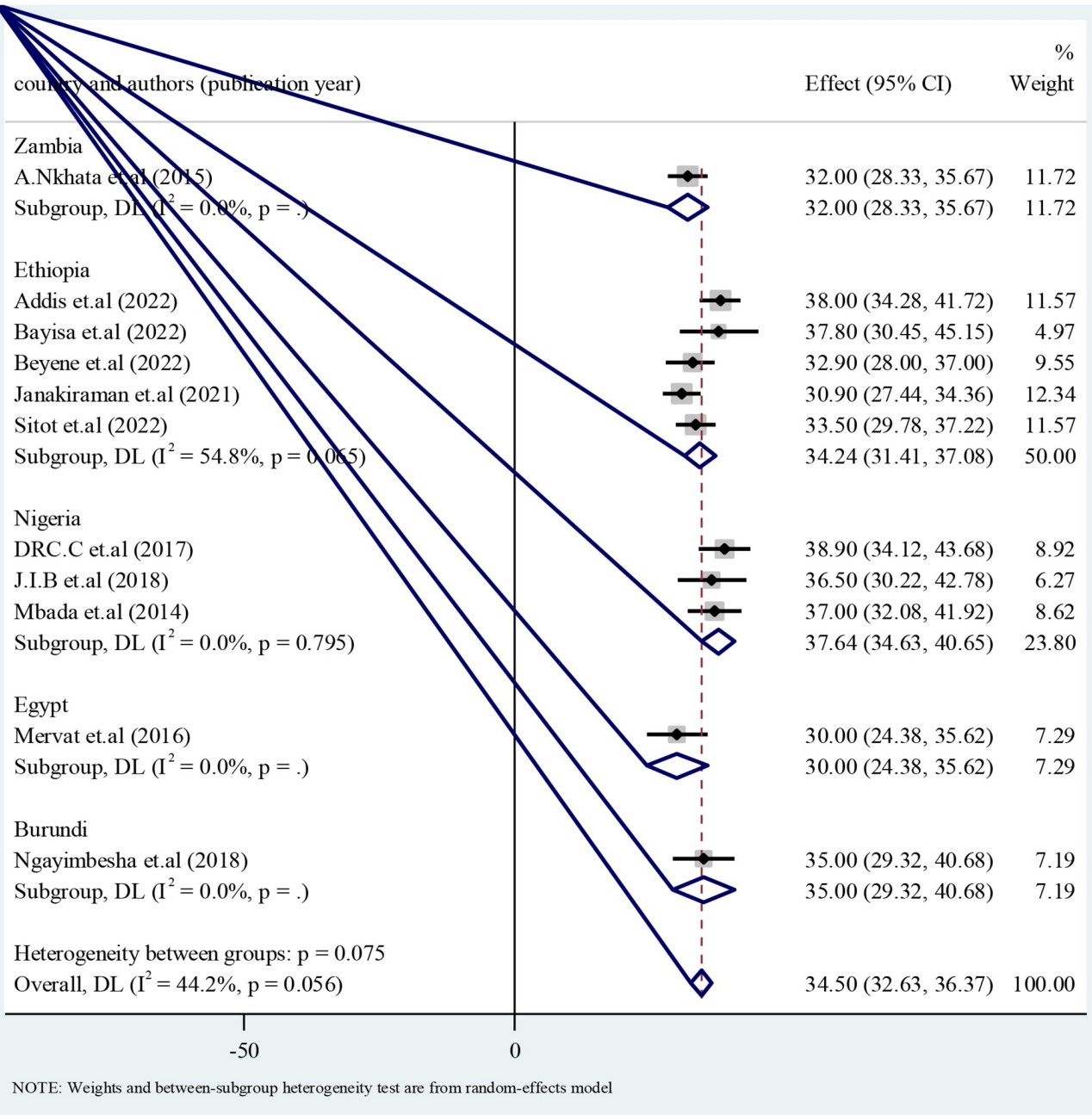

**Fig 4. Forest plot of subgroup analysis by country for pooled antenatal exercise in Africa, 2022.**

This study showed that antenatal exercise in Nigeria was 37.64 with 95% confidence interval of 34.63 to 40.65. Contrary to this a study conducted in India reports antenatal exercise practice was very much lower than Nigeria [19]. This difference might be due to most of the studies conducted in Nigeria were in the Urban area by which health care deliveries were better. In the other hand the difference might also be due oldness of the study conducted in India.

## 5. Strength and limitation

This meta-analysis brings pooled effect estimate analysis of all primary studies conducted in Africa. Antenatal exercise practice in all article were assessed to have a single pooled effect. But available studies in Africa on antenatal exercise were too limited. This may compromise representativeness of this study to the general population.

## 6. Conclusion and recommendation

The overall pooled effect estimate of antenatal exercise in Africa was low compared to other continent. There is a need for more studies to examine the dynamics of antenatal exercise in Africa to guide contextual interventions to improve and promote maternal health in Africa.

Therefore the following recommendations were given based on the obtained result of this study:

- As it was recommended by ACOG antenatal exercise to every patient in the absence of contraindications it should be encouraged by professionals providing antenatal and preconception care service.

- Larger promotion on antenatal exercise to African community may need to be provided African regional office of WHO.

- Since the available data on antenatal exercise in Africa were too small further researches need to be conducted to have more robust result.

## Supporting information

**S1 Table. PRISMA checklist.**
(DOCX)

**S2 Table. NOS quality assessment.**
(DOCX)

**S1 Dataset.**
(DTA)

**S1 File.**
(DTA)

## Acknowledgments

Authors of the primary research used on this systematic meta-analysis never need to be missed from acknowledgment.

## Author Contributions

**Conceptualization:** Yohannes Fikadu Geda.

**Data curation:** Yohannes Fikadu Geda, Seid Jemal Mohammed, Tamirat Melis Berhe, Samuel Ejeta Chibsa, Tadesse Sahle, Yirgalem Yosef Lamiso, Kenzudin Assfa Mossa, Molalegn Mesele Gesese.

**Formal analysis:** Yohannes Fikadu Geda, Seid Jemal Mohammed, Tamirat Melis Berhe, Samuel Ejeta Chibsa, Tadesse Sahle, Yirgalem Yosef Lamiso, Kenzudin Assfa Mossa, Molalegn Mesele Gesese.

**Funding acquisition:** Yohannes Fikadu Geda, Seid Jemal Mohammed, Tamirat Melis Berhe, Samuel Ejeta Chibsa, Tadesse Sahle, Yirgalem Yosef Lamiso, Kenzudin Assfa Mossa, Molalegn Mesele Gesese.

**Investigation:** Yohannes Fikadu Geda, Seid Jemal Mohammed, Tamirat Melis Berhe, Samuel Ejeta Chibsa, Tadesse Sahle, Yirgalem Yosef Lamiso, Kenzudin Assfa Mossa, Molalegn Mesele Gesese.

**Methodology:** Yohannes Fikadu Geda, Seid Jemal Mohammed, Tamirat Melis Berhe, Samuel Ejeta Chibsa, Tadesse Sahle, Yirgalem Yosef Lamiso, Kenzudin Assfa Mossa, Molalegn Mesele Gesese.

**Project administration:** Yohannes Fikadu Geda, Seid Jemal Mohammed, Tamirat Melis Berhe, Samuel Ejeta Chibsa, Tadesse Sahle, Yirgalem Yosef Lamiso, Kenzudin Assfa Mossa, Molalegn Mesele Gesese.

**Resources:** Yohannes Fikadu Geda, Seid Jemal Mohammed, Tamirat Melis Berhe, Samuel Ejeta Chibsa, Tadesse Sahle, Yirgalem Yosef Lamiso, Kenzudin Assfa Mossa, Molalegn Mesele Gesese.

**Software:** Yohannes Fikadu Geda, Seid Jemal Mohammed, Tamirat Melis Berhe, Samuel Ejeta Chibsa, Tadesse Sahle, Yirgalem Yosef Lamiso, Kenzudin Assfa Mossa, Molalegn Mesele Gesese.

**Supervision:** Yohannes Fikadu Geda, Seid Jemal Mohammed, Tamirat Melis Berhe, Samuel Ejeta Chibsa, Tadesse Sahle, Yirgalem Yosef Lamiso, Kenzudin Assfa Mossa, Molalegn Mesele Gesese.

**Validation:** Yohannes Fikadu Geda, Seid Jemal Mohammed, Tamirat Melis Berhe, Samuel Ejeta Chibsa, Tadesse Sahle, Yirgalem Yosef Lamiso, Kenzudin Assfa Mossa, Molalegn Mesele Gesese.

**Visualization:** Yohannes Fikadu Geda, Seid Jemal Mohammed, Tamirat Melis Berhe, Samuel Ejeta Chibsa, Tadesse Sahle, Yirgalem Yosef Lamiso, Kenzudin Assfa Mossa, Molalegn Mesele Gesese.

**Writing – original draft:** Yohannes Fikadu Geda.

**Writing – review & editing:** Yohannes Fikadu Geda, Seid Jemal Mohammed, Tamirat Melis Berhe, Samuel Ejeta Chibsa, Tadesse Sahle, Yirgalem Yosef Lamiso, Kenzudin Assfa Mossa, Molalegn Mesele Gesese.

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
