## [Decision Letter · Decision Letter 0]

7 Jun 2023

PONE-D-23-05963Do pregnant African women exercise? A meta-analysisPLOS ONE

Dear Dr. Geda,

Thank you for submitting your manuscript to PLOS ONE. After careful consideration, we feel that it has merit but does not fully meet PLOS ONE’s publication criteria as it currently stands. Therefore, we invite you to submit a revised version of the manuscript that addresses the points raised during the review process.

*
**An expert in the field handled your manuscript and brought in their comments to authors some very important points to strengthen your study. All of these comments must be addressed in a tracked-changed document and outlined a response to reviewers file (more about resubmission documents are bulleted below) to be considered for review. **
*

We look forward to receiving your revised manuscript.

Kind regards,

Frank T. Spradley

Academic Editor

PLOS ONE

Reviewers' comments:

Reviewer's Responses to Questions

**Comments to the Author**

1. Is the manuscript technically sound, and do the data support the conclusions?

Reviewer #1: Yes

2. Has the statistical analysis been performed appropriately and rigorously? 

Reviewer #1: Yes

3. Have the authors made all data underlying the findings in their manuscript fully available?

Reviewer #1: Yes

4. Is the manuscript presented in an intelligible fashion and written in standard English?

Reviewer #1: No

5. Review Comments to the Author

Reviewer #1: I would like to appreciate the authors for their effort to come up with this interesting subject which has paramount importance in pregnant women health. As life style gradually becoming sedentary both in developed and developing countries, assessing prevalence of exercise particularly among pregnant women is topic need to studied. I have reviewed the document thoroughly and seen the document was well done. However, some of basic things in meta-analysis study need to be Clearfield. My comments and questions that author should address are listed herewith

Methods

Study design and setting

Prospero's registration number should be indicated.

Eligibility criteria

If the current meta-analysis included all studies conducted in any African country, how have you accessed those studies published in a language other than English?

It is not easy to incorporate unpublished master’s thesis and PhD dissertations from repositories in whole African countries. So, what strategies were used to get those studies should be clarified under search strategy.

Some of the information under inclusion criteria is about study selection. So, it is better to give it a separate title and describe it with a figure.

Quality assessment

What are the criteria considered to assess the quality of the articles included in the study? It was indicated that you have used the Newcastle-Ottawa scale (NOS). But what strategy have you used to assess the scale? Is it done by one of the authors or an independent assessment? What was the score of the assessment? Is there any article of poor quality? If yes, what has been done?

Data analysis

The author declared that there was significant heterogeneity observed across studies and used a random effect model for it. It is obviously an expected event, as you are collecting information from the whole continent of Africa. Hence, analysis to understand the source of heterogeneity will be helpful. It guides the author to do subgroup analysis. So, how do you detect possible sources of heterogeneity, and have you done the subgroup analysis? Is the source of heterogeneity is the country?

I was wondering if I could see the list and definition of your study variables in the methods section, but I could not. How do you define your outcome variable "exercises during the antenatal period"? How the primary studies measured exercise during pregnancy. As we know, most pregnant women in Africa remain active even in late pregnancy due to their lifestyle. So, how do you differentiate this routine lifestyle physical activity from what we call two or three days of exercise in a week?

What instruments were used by primary studies to assess the exercise level of pregnant women?

Conclusion and recommendation

The objective/result of your study was the pooled prevalence of physical exercise among pregnant women. You have not assessed any factor associated with it, however.

You are recommending

To change policies and maternal health guidelines, and health promotion activities in Africa. How do you know these are the factors contributing to low physical activity among pregnant women? Your recommendation should be in line with your work.

lastly, grammar and punctuation of the manuscript should be rechecked

6. PLOS authors have the option to publish the peer review history of their article (what does this mean?). If published, this will include your full peer review and any attached files.

Reviewer #1: **Yes: **Teshome Gensa Geta

---

## [Author Response · Author response to Decision Letter 0]

16 Jun 2023

Dear reviewer, we are excited to incorporate your insightful criticism to improve our article. As a result, we changed the manuscript; the changes are indicated in the revised version by track changes. Below are point by point responses to the reviewer concerns.

Reviewer #1: I would like to appreciate the authors for their effort to come up with this interesting subject which has paramount importance in pregnant women health. As life style gradually becoming sedentary both in developed and developing countries, assessing prevalence of exercise particularly among pregnant women is topic need to studied. I have reviewed the document thoroughly and seen the document was well done. However, some of basic things in meta-analysis study need to be Clearfield. My comments and questions that author should address are listed herewith

Methods

Study design and setting

Prospero's registration number should be indicated.

Response: It’s mentioned in the current version of the manuscript. 

Eligibility criteria

If the current meta-analysis included all studies conducted in any African country, how have you accessed those studies published in a language other than English?

Response: Not at all; studies published in a language other than English were excluded from this study. 

It is not easy to incorporate unpublished master’s thesis and PhD dissertations from repositories in whole African countries. So, what strategies were used to get those studies should be clarified under search strategy.

Response: Search strategy of this study was revised in the current version. 

Some of the information under inclusion criteria is about study selection. So, it is better to give it a separate title and describe it with a figure. 

Response: Separate sub-section for study selection was given under the methods and materials section, and it’s already described in figure 1. 

Quality assessment

What are the criteria considered to assess the quality of the articles included in the study? It was indicated that you have used the Newcastle-Ottawa scale (NOS). But what strategy have you used to assess the scale? Is it done by one of the authors or an independent assessment? What was the score of the assessment? Is there any article of poor quality? If yes, what has been done?

Response: All articles were independently verified by the two authors (YFG and SJM). Where disagreements have occurred, the articles have been reviewed by one of the authors (TMB) and used as final mediation and admissibility decision. The NOS was included as a supplementary table 2 (S_table2) in the current submission. As it was described on the exclusion criteria studies with poor qualities like articles that did not have proven methods, articles that do not contain comprehensive information of importance for analysis and case reports and articles with inconsistent measures of outcome variables were excluded from the final analysis of this study. 

Data analysis

The author declared that there was significant heterogeneity observed across studies and used a random effect model for it. It is obviously an expected event, as you are collecting information from the whole continent of Africa. Hence, analysis to understand the source of heterogeneity will be helpful. It guides the author to do subgroup analysis. So, how do you detect possible sources of heterogeneity, and have you done the subgroup analysis? Is the source of heterogeneity is the country?

Response: Yes, in this investigation, national differences were discovered to be one of the sources of heterogeneity among the studies considered.

I was wondering if I could see the list and definition of your study variables in the methods section, but I could not. How do you define your outcome variable "exercises during the antenatal period"? How the primary studies measured exercise during pregnancy. As we know, most pregnant women in Africa remain active even in late pregnancy due to their lifestyle. So, how do you differentiate this routine lifestyle physical activity from what we call two or three days of exercise in a week?

Response: The "operational definition" was included in the methods and materials section of the most recent version of the manuscript. There can be no denying that pregnant women in rural Africa engage in strenuous physical activity such as carrying, fetching, and other tasks. Contrarily, compared to rural women, pregnant African women live opposite lifestyles in urban areas. Furthermore, it may be difficult to argue that pregnant African women living in rural areas do not require pre-planned physical activity. Due to these and the reviewer's concerns, we already recommended conducting a comparative meta-analysis.

Conclusion and recommendation

The objective/result of your study was the pooled prevalence of physical exercise among pregnant women. You have not assessed any factor associated with it, however.

You are recommending

To change policies and maternal health guidelines, and health promotion activities in Africa. How do you know these are the factors contributing to low physical activity among pregnant women? Your recommendation should be in line with your work.

Response: The recommendation was revised in the current submission. 

lastly, grammar and punctuation of the manuscript should be rechecked

Response: Grammar and punctuation of the manuscript were thoroughly revised in the current version.

Thank you!

---

## [Decision Letter · Decision Letter 1]

19 Jul 2023

Do pregnant African women exercise? A meta-analysis

PONE-D-23-05963R1

Dear Dr. Geda,

We’re pleased to inform you that your manuscript has been judged scientifically suitable for publication and will be formally accepted for publication once it meets all outstanding technical requirements.

Kind regards,

Frank T. Spradley

Academic Editor

PLOS ONE

Reviewers' comments:

Reviewer's Responses to Questions

**Comments to the Author**

1. If the authors have adequately addressed your comments raised in a previous round of review and you feel that this manuscript is now acceptable for publication, you may indicate that here to bypass the “Comments to the Author” section, enter your conflict of interest statement in the “Confidential to Editor” section, and submit your "Accept" recommendation.

Reviewer #1: All comments have been addressed

2. Is the manuscript technically sound, and do the data support the conclusions?

Reviewer #1: Yes

3. Has the statistical analysis been performed appropriately and rigorously? 

Reviewer #1: Yes

4. Have the authors made all data underlying the findings in their manuscript fully available?

Reviewer #1: Yes

5. Is the manuscript presented in an intelligible fashion and written in standard English?

Reviewer #1: Yes

6. Review Comments to the Author

Reviewer #1: The author addressed all the questions raised during review process. I recommended the authors to use standard question to assess physical activity level of pregnant mother to plan for future primary researches.

7. PLOS authors have the option to publish the peer review history of their article (what does this mean?). If published, this will include your full peer review and any attached files.

Reviewer #1: No

---

## [Editor Report · Acceptance letter]

29 Aug 2023

PONE-D-23-05963R1 

Do pregnant African women exercise? A meta-analysis 

Dear Dr. Geda:

I'm pleased to inform you that your manuscript has been deemed suitable for publication in PLOS ONE. Congratulations! Your manuscript is now with our production department. 

Kind regards, 

on behalf of

Dr. Frank T. Spradley 

Academic Editor

PLOS ONE